# Implementing Nature-Based Solutions in Urban Spaces in the Context of the Sense of Danger That Citizens May Feel

**Barbara Vojvodíková** [1,*] , **Iva Tichá** [2] and **Anna Starzewska-Sikorska** [3]

1   IURS—Institute for Sustainable Development of Settlements, 70800 Ostrava, Czech Republic
2   Faculty of Social Studies, University of Ostrava, 70200 Ostrava, Czech Republic
3   Institute for Ecology of Industrial Areas, 40-844 Katowice, Poland
*   Correspondence: barbara.vojvodikova@email.cz; Tel.: +420-725-117-244

**Abstract:** Cities are facing the challenges of climate change. The application of nature-based solutions (NBS) to the urban structure is often mentioned in climate change adaptation strategies. In an effort to ensure the greatest possible well-being of citizens in the form of environmentally positive elements, the opinions of citizens are forgotten. This paper presents the results of research focusing on the feelings of unsafety associated with the application of NBS elements directly into the urban structure. In two pilot areas (Ostrava (CZ) and the part of Upper Silesian agglomeration (PL)) the feelings of the inhabitants and the possible feeling of fear or danger in the application of NBS were investigated. In Ostrava, a questionnaire survey was conducted in relation to specific elements of the NBS without discussion of specific locations. In the Upper Silesian agglomeration, residents' feelings about specific NBS were surveyed at specific locations using guided interviews. Both approaches resulted in the identification of elements of concern. Respondents who discussed a specific location had a better understanding of the urban context and worried less. The two approaches demonstrated the need to communicate with residents before finalizing the design of a particular public space and the desirability of discussing site-specific issues with citizens.

**Keywords:** nature-based solution; worries; sustainable cities; brownfields

## 1. Introduction

Since their creation, cities have undergone a series of changes resulting from the changing needs of their inhabitants (fortification and defortification, industrialization and deindustrialization, etc.) [1]. These changes have been manifested in recent decades through densification of urban cores, consolidation of areas and expansion of cities into suburbs [2–4]. Cities are home to more than half of the world's population and are responsible for three-quarters of global energy consumption and greenhouse gas emissions [5]. Just as cities themselves are changing, [6] the perceptions and needs of city residents are also changing, which may be defined in different ways e.g., [7], but the hierarchy of residents' needs [8] and the importance of feeling safe is still relevant. In [9], it is stated that human wants contribute to making cities "richer, smarter, greener, healthier and happier".

Cities face many challenges in their pursuit of sustainable development. A much-cited challenge today is resilience to challenges associated with climate change [10–12]. The fundamental challenge is to develop cities as holistic systems that cater to human needs in the context of environmental and social justice [13]. When applying measures such as NBS, it is essential to always reduce hazards and promote well-being [14]. Climate change is leading to a gradual increase in urban temperatures, resulting in the creation of more intense heat islands [15–17]. Climate change is also increasing flood risk [18], which is mainly associated with pluvial flooding [19], causing localised damage often unrelated to the overtopping of a permanent water body. The application of green elements has become a key strategy for municipalities in adapting to the climate effect of the urban heat island,

stormwater runoff and water pollution, as well as an appropriate solution to improve the visual image of the city [20].

Cities also have land in their structure that is contaminated mainly by previous industrial activities. Soil contamination has a strong environmental impact. For example, phytoremediation, which uses the absorption, accumulation and decomposition of contaminants, can be applied to reduce or limit the mobility of contaminants [21]. When considering the application of a green feature in areas where contamination is confirmed, it is advisable to select suitable plants in terms of their phytoremediation capacity.

Proper management of urban green spaces and their well-planned structure has a positive impact on both reducing heat islands and the consequences of pluvial flooding [22,23]. In urban areas, nature-based solutions play an important role [24]. Many cities have parks and green spaces larger than 2 ha in their urban fabric [25,26]. However, these green spaces are usually quite far apart, and their influence does not cover the entire area of the city [27]. In a dense urban structure, it is rarely possible to introduce new green spaces of large sizes, such as parks, and other solutions must be sought [28,29]. One such solution may be the introduction of small green spaces, for which the name "urban environmental acupuncture" has recently been adopted and which is closely related to nature-based solutions (NBS).

Several requirements for the integration of green spaces into the urban fabric can be found in adaptation strategies [30–35]. The negotiation of individual implementation measures of adaptation strategies responding to climate change faces a number of obstacles at the municipal level [36]. In addition to environmental aspects, an economic effect is also associated with the application of NBS. Especially in the case of established parks and urban forests, the influence of the increase in real estate prices [14], the occupancy of housing and the increase in the well-being of the territory [21] are mentioned.

One aspect that influences the adoption of an NBS element is the feeling of safety or the feeling of danger and threat. The relationship between greenness in the city and fear of danger was already known in the 19th century [37]. According to [8], safety is considered the second most basic human need. Feeling safe refers to a certain level of security perceived by an individual [38].

The level of safety is closely related to the perception of safety in the place where the inhabitants are. Since public green space has been mentioned in relation to pedestrian movement, it is essential to create a safe space for pedestrians [39,40]. Projects including safe streets for improving pedestrian include not only traffic infrastructure but also the renovation of smaller green or open spaces [41].

Places that are usually associated with negative emotions —most commonly with fear [42,43]—as a result of crime or criminal activity [44–47] are those that provide potential hiding places for perpetrators, limit visibility or the ability to escape [48].

Fear can also be associated with various types of phobias. For example, general fear of animals, fear of insects and fear of birds. In general, some people may suffer from zoophobia (zoophobia) fear of animals [49]. Entomophobia (the fear of insects) can be also associated with allergies or traumatic experiences with insects [50]). The flowering elements of NBS may be surrounded by pollinating insects. Should residents suffer from ornithophobia (fear of birds), they might feel threatened if a bird species nests in an element [51].

The aim of the research was to find out which elements of NBS are perceived as dangerous (pilot area in Ostrava) and whether personal experience and active consultation can influence these perceptions (part of Upper Silesian agglomeration in Poland) so that their application by city representatives is best received by residents. The research was based on the assumption that individual NBS elements would be placed on individual sites in accordance with the definition prepared by the Salute4CE project where a potentially suitable place for the application of a nature-based solution is:

- a place that is not maintained, is neglected, or no longer fulfils its function (brownfields);
- a smaller site—ideally up to 0.2 ha but not more than 0.6 ha to allow feasible implementation; or

- a place that spoils the image of its surroundings or even reduces property prices in its vicinity.

Brownfields present an opportunity for new development projects. They are also sites for public green space. In special cases, linear brownfields (railway brownfields) can be used as bio-corridors [52]. Brownfields appear in some research to be very suitable, for example, for the location of urban forests, where they bring cooling effects and increase biodiversity [53]. By the loss of their original (e.g., industrial) function, they are suitable sites for change and the location of some elements of the NBS. These elements can also serve to enhance the process of phytoremediation [21]. However, the research presented in this article was focused on brownfields up to 0.6 ha. This practically excluded former light or heavy industry enterprises which are characterized by a high level of contamination.

Two areas in the Upper Silesian Coal Basin were selected for research: Ostrava (CZ) and part of Upper Silesian agglomeration (PL).

## 2. Pilot Areas in Upper Silesian Coal Basin

The territory is spread over the area of Poland and the Czech Republic, which are connected by common history of coal mining and heavy industry. For the Czech part, Ostrava as a metropolitan city was chosen and for the Polish part, was selected three cities Chorzów, Ruda Śląska and Świętochłowice (part of Upper Silesian agglomeration).

### 2.1. Pilot Area—Ostrava (CZ)

The city of Ostrava is part of the Moravian-Silesian Region in the Czech Republic. It has 279,791 (as of 1 January 2022) [54] inhabitants. The city is rich in industrial history—coal mining began in 1787 and ended in 1993 [55], and in 1828, the ironworks in Vítkovice (now one of Ostrava's districts) were founded (part of it still operates today). Ostrava is a typical shrinking city, with an ageing population [56]. In terms of the environment, Ostrava was one of the worst polluted areas in the Czech Republic. Thanks to the decline of coal mining, industrial restructuring and investments aimed at improving Ostrava's environment, the environment and air quality have gradually improved. Nevertheless, in recent years, according to the measurements of the Czech Hydrometeorological Institute, it has become one of the most polluted cities in the Czech Republic in terms of pollution by the carcinogen benzopyrene. The concentrations of dust particles in Ostrava's air have also scored the highest rank in the country [57].

The city tries to maintain larger areas of green space, which is achieved, among other things, thanks to its loose urban structure [58]. The City of Ostrava manages the green spaces located on city-owned land in 23 urban districts. At the end of 2019, the total area of green space on city-owned land (excluding forest areas) was approximately 1867 ha. In the urban district of Moravská Ostrava and Přívoz (city centre), this amounts to 225 ha (most of which is located in Moravská Ostrava). In the administrative district of the municipality of Ostrava (the city itself and several surrounding municipalities), there are currently 5399.9 ha of forests, mostly deciduous and mixed [59].

The city of Ostrava is trying to behave responsibly when dealing with the challenges of the standard of living, air pollution, and climate change. It has therefore developed the Strategic Development Plan of the City of Ostrava 2017–2023 and the Adaptation Strategy—Healthy Ostrava [60].

The Adaptation Strategy 2017–2023 notes that a gradual increase in average annual temperature by approximately 2.5–3 °C (compared to the average for the period 1961–2009), more frequent occurrence of heat waves and an increase in their duration can be expected by 2100. By 2100, can be expected a significant increase in summer temperatures by 3–4 °C and therefore a significant increase in the number of tropical days and a decrease in the number of arctic days. The most affected localities mentioned in the text are the urban districts of Vítkovice, Moravská Ostrava and Přívoz, and Hrabůvka. For the preparation of the Adaptation Strategy, the City of Ostrava organised a mapping survey where residents marked their perceived thermal discomfort on a map. (In the text, the term thermal

discomfort is used if the authors are referring to sensible heat. Thermal stress is used in the context of heat and overheating problems that have been measured or modelled).

The Ostrava City Development Strategy [61] also addresses feelings of safety as part of the standard of living. Therefore, the strategy includes an emotional map of places where residents do not feel safe.

As described above, NBS applications should be dominated by sites that are inappropriately used, abandoned, or forgotten. No such inventory of sites exists for the city of Ostrava. Therefore, the authors of the article used the database of brownfield sites with the knowledge that regeneration of these areas will not be primarily directed towards the use of NBS; however, with the current setup of the whole society, it is possible to assume the application of some NBS as an integral part of the regeneration process.

For the territory of the city of Ostrava, the brownfields database is managed by MSID (a company owned by the region).

Brownfields are defined for the purposes of the database as sites or buildings that have been affected by previous use. Now, they have lost their function and are only partially used or even completely abandoned. They need some necessary support for further or different use [62].

There are 122 brownfields registered in Ostrava. Of these brownfields, 83 sites are smaller or equal to 0.6 ha. Of these 83 sites, 62 are smaller than 0.2 ha. All these sites are listed as sites with no proven contamination. In terms of their past use, these are mostly areas previously used for housing, office space or some small transport structures such as bus stops or garages. The database also includes an assessment of their potential for future use. The authors are aware that this information is not directly related to the application of the NBS on these sites, but it illustrates the overall situation.

To visualize the whole situation, a map of brownfields was linked to maps of perceived temperature discomfort (Figure 1) and perceived danger (Figure 2).

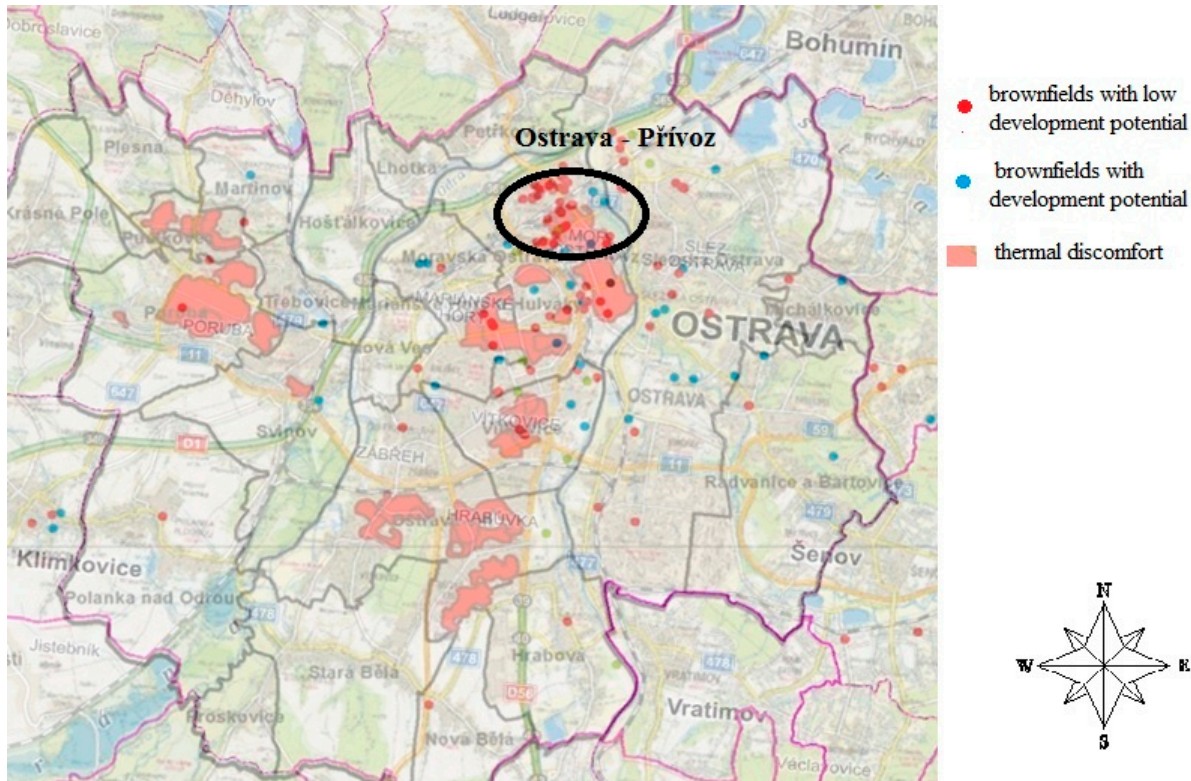

**Figure 1.** Connecting brownfield sites and places where the residents of Ostrava feel thermal discomfort. (source: based on information from [59,60,62] created by authors).

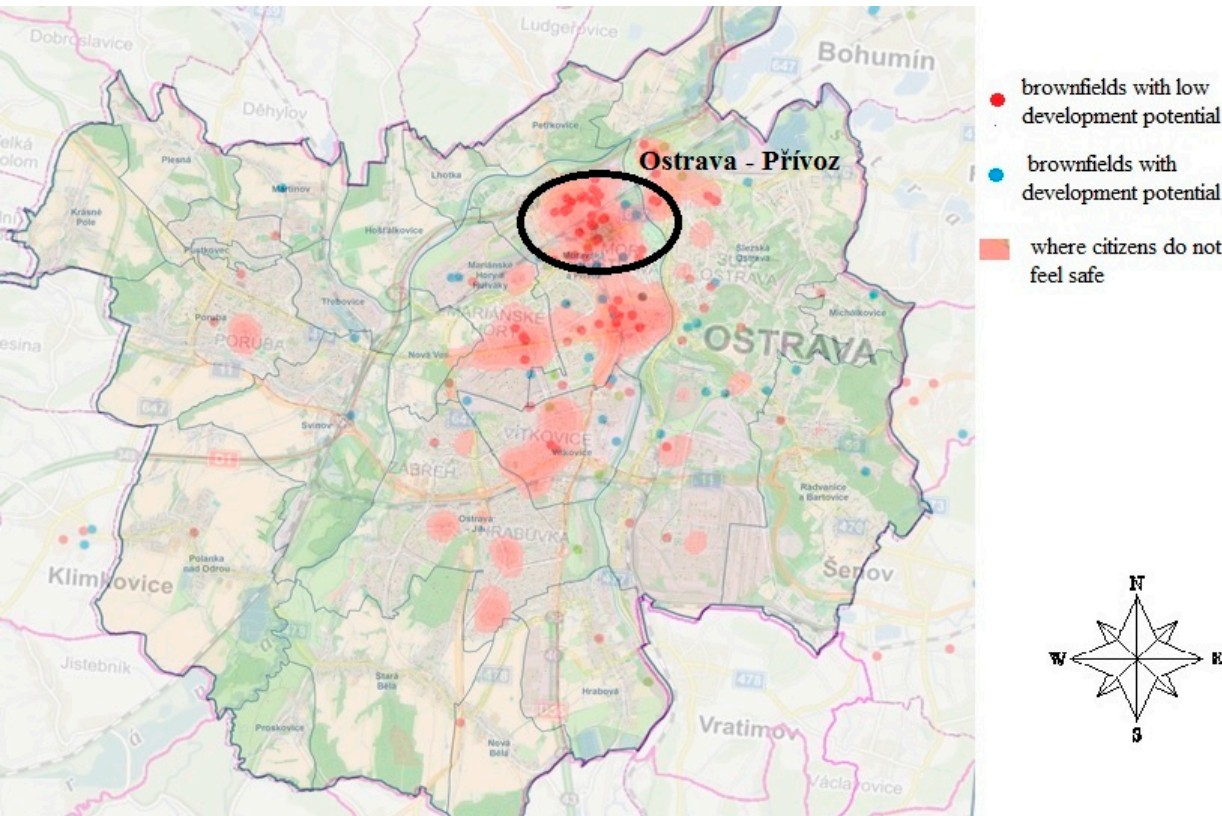

**Figure 2.** Connecting brownfield sites and places where the residents of Ostrava feel danger. (source: based on information from [59,60,62] created by authors).

The authors are aware that the perception of thermal discomfort is influenced by a number of personal and/or physical characteristics [63,64] and differs from measurement-based temperature maps. For elderly residents, heat can be a health risk or even fatal. [65,66] Similarly, hazard perception and crime maps differ.

Given the targeting of feelings and perceptions of safety towards elements of the NBS, the authors consider the feeling maps to be more appropriate and illustrative.

When linking the feeling of thermal discomfort maps to the brownfields map (Figures 1 and 2), it is clear that there is a high concentration of brownfields (marked by dots) and a feeling of thermal discomfort, especially in the Přívoz district. The same problem site can be observed in the connection between the brownfields map and the feeling of danger (not feeling safe).

### 2.2. Pilot Area—The Part of Upper Silesian Agglomeration(PL)

Functional Urban Area (FUA) represents three cities: Chorzów, Ruda Śląska and Świętochłowice located within the Metropolitan Association of Upper Silesia and Dąbrowa Basin, usually referred to in Poland as the Silesian Metropolis or Metropolis GZM. The Silesian Metropolis consists of 41 municipalities in the Silesian Voivodeship in Poland. The FUA covers an area of 124.19 km$^2$ with over 250,000 inhabitants and is located in the centre of Metropolis GZM, which is very important for the development of the area. The peculiar structure of land use is mainly due to the many years of industrial operation within its borders. The industrial landscape was characterised by industrial and post-industrial buildings and areas located in the immediate neighbourhood of city centres as well as by spoil heaps and dumping sites. The proportion of anthropogenic areas reaches over 55% of the whole FUA surface, implying a high level of its transformation.

In the area, two main challenges have been indicated as related to climate change urban resilience: heat island and soil sealing (The core of this article focuses on the problem of thermal stress).

The analysis of land surface temperature (LST) showed the LST distribution in Chorzów, Świętochłowice, and Ruda Śląska for the warm seasons period 2016–2020 [50]. After collecting and compiling the statistical data, it can be concluded that the highest maximum temperatures (above 39 °C) were recorded in Chorzów City, which is characterised by a dense urban fabric. The city of Świętochłowice is characterised by the highest average temperature (above 27 °C), which is caused by the smallest area of the city and the smallest surface of green areas (including forests). Due to the lack of green spaces and forests, Świętochłowice has the highest minimum surface temperature of above 22 °C among the analysed cities. The city of Ruda Śląska, characterised by separate districts and a polycentric structure, recorded the lowest minimum temperature of 21.11 °C, and the lowest mean temperature of 26.07 °C. In the entire analysed area, over 60% of the surface is characterised by a temperature of 25–30 °C. 22% of the area of Świętochłowice and 26% of the area of Chorzów have an average temperature below 25 °C, 10% of the area of Chorzów and Świętochłowice has a temperature of 30–35 °C. The highest percentage of the city area with a temperature above 35 °C was observed in Chorzów (1%), and approximately 0.5% of the area of Świętochłowice and Ruda Śląska [67].

## 3. Methods

In Ostrava, the research focused on feelings of danger of implementing the NBS without targeting the application to a specific location. Respondents, as described below, expressed themselves based on their feelings without having to look for solutions in their neighbourhood.

In the Silesian agglomeration, respondents were addressing site-specific NBS proposals. As part of proposing specific NBS solutions, structured interviews were conducted to determine if they felt there was any danger in applying a particular element

### 3.1. Methods Applied in Pilote Area Ostrava

3.1.1. Selection of Respondents and Communication Techniques

Three groups of respondents were selected for the questionnaire survey in Ostrava.

- Group A—respondents living in housing estates in Ostrava. The group consisted mostly of residents of retirement age. In the Czech Republic, almost 20% [68], of the population is aged 65+. (In this group, there were 3 respondents aged 65–70 and 5 respondents aged 80–85. The other respondents were in the 70–75 and 75–80 age groups). This is a significantly large group of people who tend to be more concerned about crime (analysis).

- Group B—respondents were residents of Ostrava, regardless of the urban district and type of housing, but their age was up to 26 years. (All respondents in this group were aged 20–26). This group of respondents was composed mainly of students. This age group is the least worried about crime [69], but at the same time, they feel very strongly about the problems associated with climate change.

- Group C—respondents were residents of urban districts with a higher crime rate, including residents of the Přívoz district. Respondent age ranged from 26 to 60 years. (The age structure of respondents in this group was: 2 respondents 26–30 years, 4 respondents 35–40 years, 3 respondents 40–45 years, 5 respondents 45–50 years, 1 respondent 55–60 years and 1 respondent 60–65 years.) The reason for the selection was that the area consists of a large number of brownfield sites, with a high perceived crime rate, and is also a place of perceived thermal discomfort (see Figures 1 and 2).

The selection of groups was not random. From the beginning of the research, there was interest in three groups of respondents. The aim was to focus on respondents affected by a location that is problematic (Ostrava-Přívoz). Respondents have experience with moving in public spaces that they do not consider safe. Another significant group was considered to

be the elderly population who use public spaces throughout the day and at the same time have more concerns. The younger generation was considered to be more environmentally focused. The students were deliberately chosen as they are a homogeneous group in terms of age and education. During the implementation, the presentation and questioning format had to be modified due to the COVID-19 pandemic.

Group A—respondents were asked to complete the questionnaire in person at several meetings. These meetings were organised as part of cultural and social activities for the elderly and their companions. Prior to completing the questionnaire, an introduction to the different types of NBS was given, focusing mainly on a technical description and the NBS impacts on climate change issues. The presenters did not mention anything related to safety or feeling unsafe. The questionnaire was printed in colour. The font size was adapted to this age group. Not all the participants of these presentations were willing to complete the questionnaire.

Group B—this group was approached through their lecturers at the university. These respondents were provided with an online lecture describing each element of the NBS before completing a questionnaire. The presentation included a technical description. Respondents completed the online questionnaire using Google forms.

Group C—reaching this group was the most difficult. The outreach was done through personal contacts and with the help of NGOs operating in the locality. In addition, owners of some brownfields in Ostrava-Přívoz were also asked to cooperate through MSID. Respondents received questionnaires by e-mail or printed out. They had a spoken presentation on google drive where they could learn about the different elements of the NBS.

3.1.2. Questionnaire Structure and Method of Processing

A total of 28 types of suitable NBS were selected for the survey. These NBS were selected based on a search of existing solutions [70–81]. The list was also based on the NBS identified in the SALUTE4CE project. Their list is presented in Figure 3. For better clarity, the NBS were categorized by position (horizontal, vertical, indifferent) and shape (point, line, area).

At the beginning of the questionnaire, the purpose of the survey was briefly explained. Two personal questions were inserted for further processing (gender (female/male/other), age category). The key questions in the questionnaire focused on respondents' feelings about the potential threat of a particular NBS (feeling of danger/neutral feeling/feeling of safety/no comments). The order of individual NBS in the questionnaire was set so that it started with simple horizontal elements such as a lawn. In the middle part of the questionnaire, the NBS was mentioned in relation to rainwater, for example, a rain garden. At the end of the questionnaire, there were the NBS requiring construction solutions placed, for example, on a green wall. A photograph was included with each NBS for illustration. For an example of a question, see Figure 4. Respondents were always given an opportunity to comment with their opinion.

For the self-assessment, respondents' answers were converted into points. In the case of feeling of danger, NBS was rated 3, in the case of neutral feeling, it was rated 2, and if respondents felt safe about it, it scored 1. If respondents had no opinion, NBS was rated 0. The scores from each respondent for each NBS were summed and divided by the number of non-zero responses.

*3.2. Methods Applied in Pilot Area—The Part of Upper Silesian Agglomeration*

The assessment of feelings of danger in relation to individual NBS for the cities of Chorzów, Ruda Śląska and Świętochłowice (Katowice agglomeration) was linked to public involvement in the search for concrete solutions. The direct involvement of the public was a necessary condition and at the same time enabled the application of structured interviews as one of the research methods [82].

| | | position | | |
|---|---|---|---|---|
| | | **horizontal** | **indifferent** | **vertical** |
| **largenerss/shape** | **area** | Urban meadows | Urban wilderness / succession area | Park trees |
| | | Ground cover plants | | Green facades with climbing plants |
| | | Lawn | | Wall-mounted living walls |
| | | Rain gardens (under-drained) | | Hydroponic mobile living walls / vertical gardens |
| | | Rockery | | |
| | | Ground crops of vegetables / herbs | | Hanging wall planters (as green street furniture) |
| | | Green roof /roof terrace | | |
| | **line** | Green pavements | | Street trees |
| | | | | Large shrubs |
| | | | | Linear wetlands for stormwater filtration |
| | | Road-side swales for retention and infiltration | | Hedge/hedgerow |
| | **point** | Verges / flower beds with native perennials | VRSS slopes with green fences | Fruit trees/ shrubs/ |
| | | Green pergolas/ green arbors | | Herb spiral |
| | | Street planters (as green street furniture) | Rain gardens in planter (=self-contained) | Vertical vegetable / herb gardens |
| | | Green covering shelters | | |

**Figure 3.** Selection of nature-based solutions defined by the SALUTE4CE project (source: created by authors).

1. *Lawn – select only ONE option*

1. Trávník – vyberte pouze JEDNU možnost

| pocit nebezpečí | ○ | *feeling of danger* |
| neutrální pocit | ○ | *neutral feeling* |
| pocit bezpečí | ○ | *feeling of safety* |
| nemám názor | ○ | *I have no opinion* |

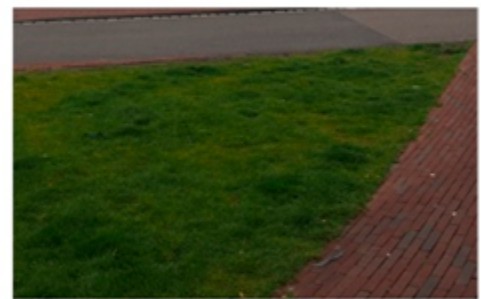

komentář k výběru Vaší odpovědi ...............................................................

, *Comment on your answer*

**Figure 4.** Questionnaire—a sample question, (English translation inserted for the purposes of this article) (source: create by authors).

Respondents were involved in planning, building, maintaining, and monitoring sites for the NBS application. Respondents were representatives of the local community—the

stakeholders. Different types of institutions and organisations were invited to participate in the discussions in the so-called living labs. The living labs were organised at the local level together with citizens and relevant experts. The aim of the living labs was to gain insights from local stakeholders (residents, civic associations, experts), to identify priorities and gain insights on bottom-up needs and opportunities in line with the ideas of public participation, while being in direct contact with respondents [83].

Discussion walks, workshops and a consultation point were organised for respondents.

Respondents—the locals themselves suggested some solutions in concrete localities and commented on feelings of safety.

## 4. Results

In general, the application of green elements in the city affects the quality of life. A number of authors, e.g., in the book *The green city and social justice* [84], mention a number of examples from the USA or Great Britain, where the expansion, modification and improvement of green areas undoubtedly led to an improvement in the quality of life and even gentrification. However, the following results show that green elements in different forms can be perceived differently.

### 4.1. Results in Pilot Area Ostrava

A total of 58 respondents provided answers for all 3 groups. The minimum number of included (not 0 or missing) answers about each individual NBS was 50. Most questions were answered by 52–55 respondents.

A total of 23 respondents from Group A were willing to fill in the questionnaire. The group of young respondents (Group B) consisted of a total of 19 respondents. Group C (from Ostrava-Přívoz) consisted of 16 respondents. (Figure 5).

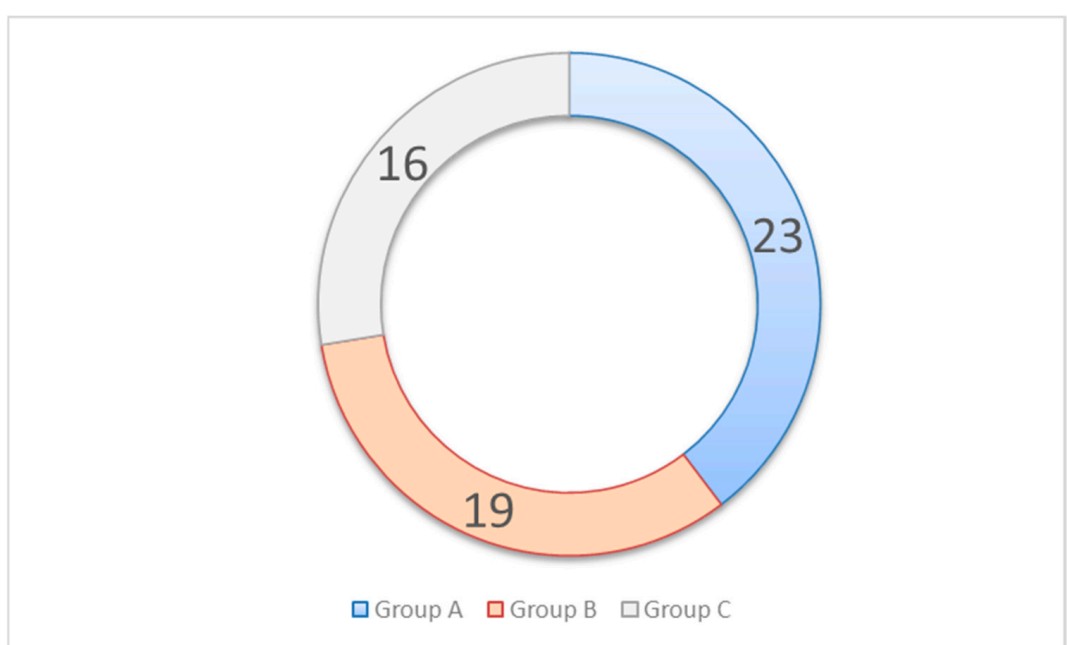

**Figure 5.** Groups with numbers of respondents (source: created by authors).

For the evaluation, the research was focused mainly on those types of natural-based solutions that have an impact on reducing the heat load. The selection was made on the basis of information [70–81] and the individual NBS were divided into the following groups according to the effect on thermal stress. The breakdown is given in Table 1.

**Table 1.** The individual NBS were divided into the following groups according to their effect on thermal stress.

| Dominant Effect | Partial Effect | Additional Effect (Has Another Dominant Function) |
|---|---|---|
| Street trees | Urban meadows | Rain gardens (under-drained) |
| Park trees | Verges/flower beds with native perennials | Road-side swales for retention and infiltration |
| Fruit trees/shrubs/ | Ground cover plants | Linear wetlands for stormwater filtration |
| Large shrubs | Lawn | Rockery |
| Hedge/hedgerow | Green pavements | Herb spiral |
| Green pergolas/green arbours | | Ground crops of vegetables/herbs |
| Green facades with climbing plants | VRSS slopes with green fences | Rain gardens in planter (=self-contained) |
| Wall-mounted living walls | Green covering shelters | Street planters (as green street furniture) |
| Hydroponic mobile living walls/vertical gardens | | |
| Vertical vegetable/herb gardens | | |
| Hanging wall planters (as green street furniture) | | |
| Green roof/roof terrace | | |
| Urban wilderness/succession area | | |

Source: authors.

The first group was a group with a dominant influence. This means that the application of the nature-based solution from the first group should reduce thermal stress. Given that this is primarily a division that will serve as a basis for structured interviews, it has been abstracted from some imperfections. For example, the effect that the tree crown size had on the efficiency of reduction of heat stress. Similarly, the height of the green wall and the type of plants used were not addressed. In the green roof solution, the distinction between intensive and extensive was not drawn.

The second group was designated as a partial influence group. These are the NBS, which have other functions such as soil treatment, soil sealing, or infiltration support. These NBS have a smaller effect on the reduction of thermal stress itself.

The third group were the NBS, which predominantly have a different purpose, especially those that support water infiltration. Even these can partially contribute to the reduction of thermal stress, but only marginally (e.g., evaporation from a rain garden).

The evaluation showed that opinions on the same type might differ. The results of the questionnaire survey in Ostrava are shown in Figure 6.

For example, young people (Group B) felt less fear, especially with NBS associated with the possibility of growing vegetables, whereas Group A and C answered mostly two or three about vertical vegetable/herb gardens, indicating that they felt neutral or even unsafe. In the group of young people of the same type, the answers were predominantly two and one, indicating that younger people perceive this element as neutral or even safe.

Another NBS type with an interesting result was a pergola. For the pergola, 15 out of 16 respondents from Group C gave a rating of three, indicating a feeling of danger, which was again consistent with Group A's answers. In Group B, a response of two (neutral) was much more common.

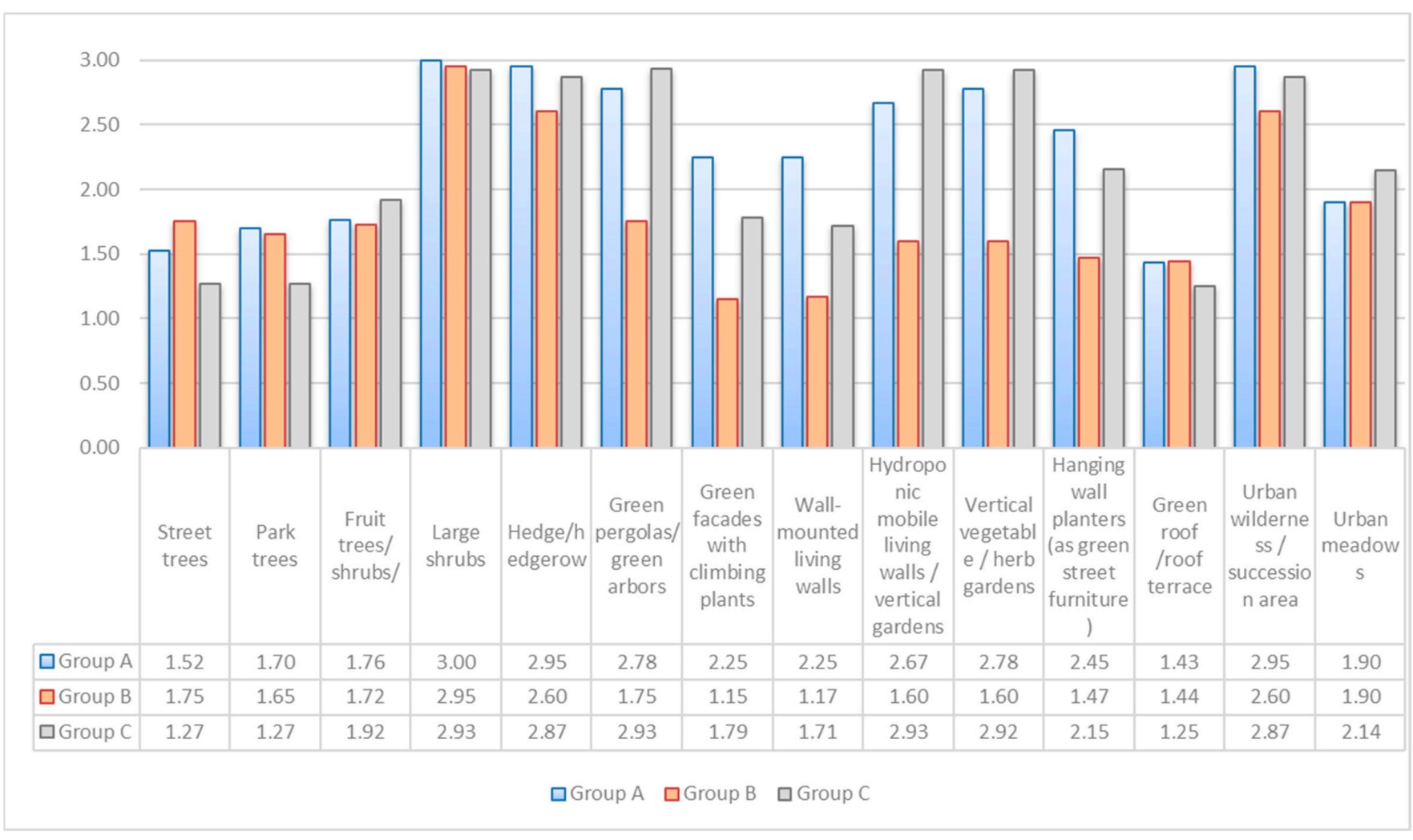

**Figure 6.** Evaluation of the feeling of danger associated with the NBS application (points: maximum 3, minimum 1) (source: created by authors).



Overall, there was a consensus on the danger posed by urban wildlife, large shrubs and hedges, with respondents from all groups expressing feelings of danger associated with these NBS. In all cases, these are visual barriers. Technical features, i.e., green facades and green walls on existing buildings, were perceived negatively for Group C, whereas green roofs were rated neutrally or positively by all. Urban meadows were considered less safe; respondents showed concerns about ticks or allergies to grass pollen.

The results also showed Group B concerns less (compared with Group A) with technical solutions—see Figure 7.

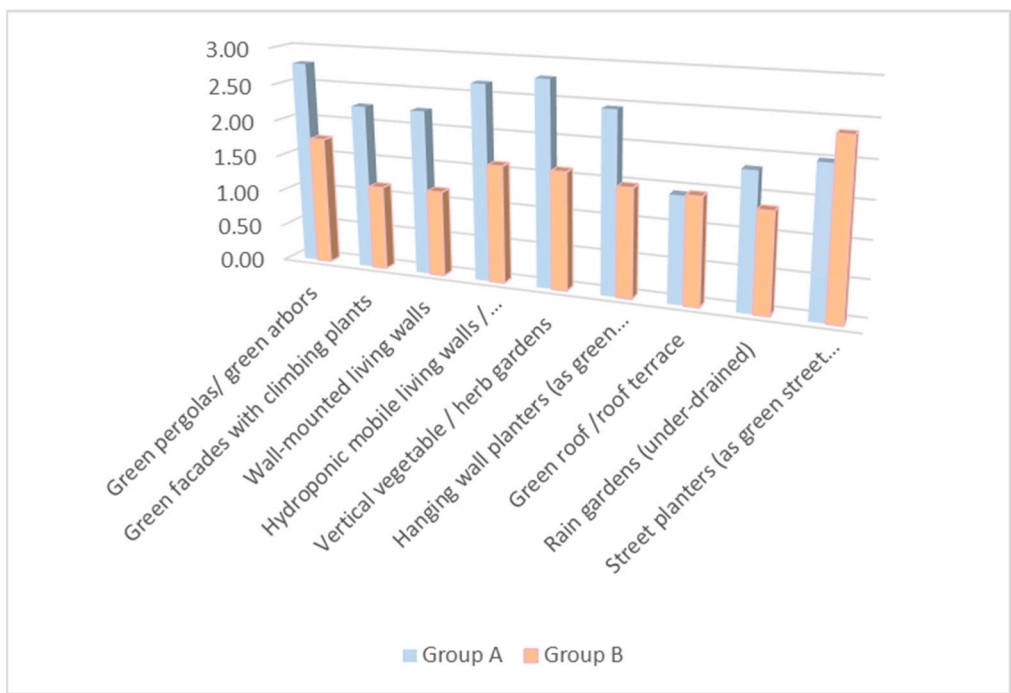

**Figure 7.** Feelings of respondents (Group A and B) focusing on NBS of technical basis. Source: authors.

Respondents were given the opportunity to comment on their answers. Table 2 shows the most recurrent comments on some types of NBS. These comments, together with the scores, are the basis for comparison with the situation in Poland.

The authors of the evaluation made minor corrections in the style of the answers, leaving the substance unchanged. Comments in the table are only given for NBS with a dominant influence on the temperature of the area.

For some types of NBS, e.g., those designed to hold water, people commented that they were concerned that a child would fall into them and drown. Despite being told what to evaluate, respondents deviated from the essence of the issue, as seen in the example of vertical vegetable/herb gardens. Respondents were concerned about damage to the NBS itself or expressed concern about liability for removing fallen leaves.

The assessment showed that respondents were concerned about those NBS that form a visual barrier, as there may be hidden threats behind this barrier. Another significant concern was the provision of shelters for drug users. Respondents are concerned that hidden locations may also serve as toilets. Nor do they prefer any features that would provide a shelter or gathering place for the homeless.

*4.2. Results in Pilot Area—The Part of Upper Silesia Agglomeration*

As described in Methods (Section 3.2), respondents considered specific locations and defined appropriate NBS solutions for them. Individual proposals were consulted with experts. Table 3 shows the respondents' comments, which were noted during the consultations (workshops, living labs etc.) (respondents always consulted on the types of

NBS in the context of a specific site). Respondents added their feelings about a particular element of the NBS. In total, 48 respondents participated in the events.

**Table 2.** Respondents' comments regarding different types of NBS.

| Types of NBS | Comments |
| --- | --- |
| Street trees | Leaves fall from the trees, and it is not clear who will take care of them, but I do not feel any danger. |
| Park trees | Leaves fall from the trees, and it is not clear who will take care of them. Will junkies gather here? |
| Fruit trees/shrubs | Children would climb trees and injure themselves. |
| Large shrubs | In no case, someone can hide behind a shrub, or it will be used as a toilet. |
| Hedge/hedgerow | There may be a thief or worse hidden behind it. |
| Green pergolas/green arbours | There will be toilets, and homeless people will gather around. |
| Green facades with climbing plants | I am worried about the technical solution so that it does not fall on anyone |
| Wall-mounted living walls | I am worried about the technical solution so that it does not fall on anyone. |
| Hydroponic mobile living walls/vertical gardens | I am worried about the technical solution so that it does not fall on anyone. |
| Vertical vegetable/herb gardens | Anyway, someone will destroy it right away and what will grow there will be stolen. |
| Hanging wall planters (as green street furniture) | It will not last long before someone destroys it. |
| Green roof/roof terrace | I'I am afraid the roof is not able to carry it. |
| Urban wilderness/succession area | Junkies would gather there. |

Source: authors.

**Table 3.** The list of results is in the part of Upper Silesian agglomeration.

| Type of NBS | Average Value for Ostrava | Respondent Comments of the Part of Upper Silesian Agglomeration |
| --- | --- | --- |
| Large shrubs | 2.959524 | Large bushes are part of, for example, Piastowski Square, forming a noise barrier. Its concept is close to urban wilderness. On this particular site, residents would welcome the retention of shrubs, but landscaped. Some respondents consider the placement of new large shrubs in a vacant gap in the development, for example, to be inappropriate concerns to raise. They are also concerned about large shrubs in a park area and near playgrounds. |
| Hedge/hedgerow | 2.806349 | A group of respondents on a particular street (11. November Street) were negative about putting trees on the street because they were concerned about overshadowing but would welcome a green fence which they did not consider risky. A green fence would separate the road from the pedestrian area. |
| Urban wilderness/succession area | 2.806349 | Respondents generally understood the importance of such an element for biodiversity and other positive attributes. The NIMBY syndrome was evident here. For no particular site, this NBS was proposed by residents who were concerned about creating space for troubled citizens. |
| Green pergolas/green arbours | 2.488647 | Respondents suggested pergolas for the courtyard area. All yards were private or semi-private spaces. Respondents did not feel apprehensive about pergolas in the courtyards. No pergola was proposed for a purely public space. |

**Table 3.** *Cont.*

| Type of NBS | Average Value for Ostrava | Respondent Comments of the Part of Upper Silesian Agglomeration |
|---|---|---|
| Vertical vegetable/herb gardens | 2.435229 | This NBS was proposed by respondents for a backyard area. All yards were either a private or semi-private space. As none of the sites in the public space discussed were suitable for growing vegetables, this NBS was not discussed by respondents from a safety perspective. |
| Hydroponic mobile (flexible) living walls/vertical gardens | 2.398413 | Respondents were positive about mobile living walls. In a few places, they even suggested them as a barrier to traffic. They felt no fear in relation to these NBS. |
| Hanging wall planters (as green street furniture) | 2.027359 | This NBS was proposed by respondents for a backyard area. All yards were either a private or semi-private space. Because none of the locations in the public space discussed were suitable for growing vegetables or herbs, this NBS was not discussed with respondents from a safety perspective |
| Urban meadows | 1.980952 | In several places, respondents suggested this type of NBS. Primarily, these were spaces today consisting of only a lawn. Often in front of shopping centres or as part of a square or park. |
| Fruit trees/shrubs | 1.800265 | Fruit trees were not proposed by respondents for any of the selected sites. Therefore, this NBS was not discussed with the respondents in terms of their safety. |
| Green facades with climbing plants | 1.728571 | Respondents in several places suggested a green facade as an appropriate solution. They felt no fear in connection with this element. |
| Wall-mounted living walls | 1.710317 | Respondents suggested this NBS for several places as an appropriate solution. They have no worries about this element. |
| Park trees | 1.53744 | A willow tree was suggested as an appropriate tree by several respondents. Willow is considered by respondents to be a very decorative tree that is also a native species. They have no fear of allergens. |
| Street trees | 1.512802 | Respondents (as indicated for the large bush section) do not want to have trees planted in their streets because they do not want to reduce the sunshine in their homes. They have no negative feelings about trees in terms of danger. |
| Green roof/roof terrace | 1.376409 | According to respondents, green roofs are very costly; therefore, they have not been proposed for any site. However, respondents reported a positive experience with green roofs in the agglomeration area. |

Source: authors.

Table 3 shows only those elements that have been evaluated for Ostrava. The green colour symbolises an agreement in the perception of the inhabitants of both pilot areas. The red colour symbolises a significant disagreement between the pilot areas. The yellow colour represents a partial disagreement. Grey elements were not proposed by respondents in the Upper Silesian agglomeration; therefore, their perception of danger was also not discussed.

For the NBS, they were evaluated by the Upper Silesian agglomeration pilot area. Graphs of response frequencies were prepared for each group of respondents in the pilot area of Ostrava. For the pilot area in Poland, unfortunately, detailed age composition is not available. The authors estimate (based on personal experience from these interviews) that this was an age group predominantly in the older working age and younger seniors. The age structure would be most closely related to Group A and Group C. According to Figures 8–10, the difference in the attitude of the young (Group B) is evident. It is also clear from Figure 9 that Group C is just strongly influenced by the negative experience of the area that is problematic. In the pilot area in Poland, none of the sites were left in such a significantly negative environment. This is an important experience for further research.

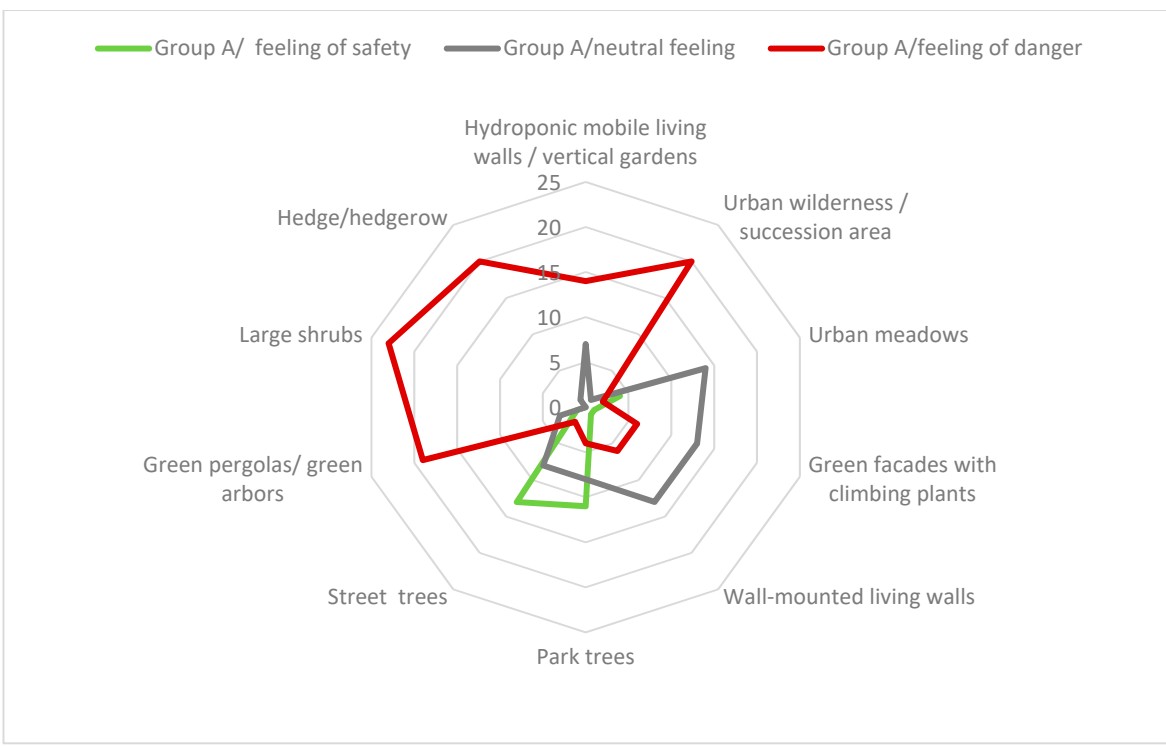

**Figure 8.** Radar charts with frequency of answers for selected NBS for the 65+ group Source: authors.

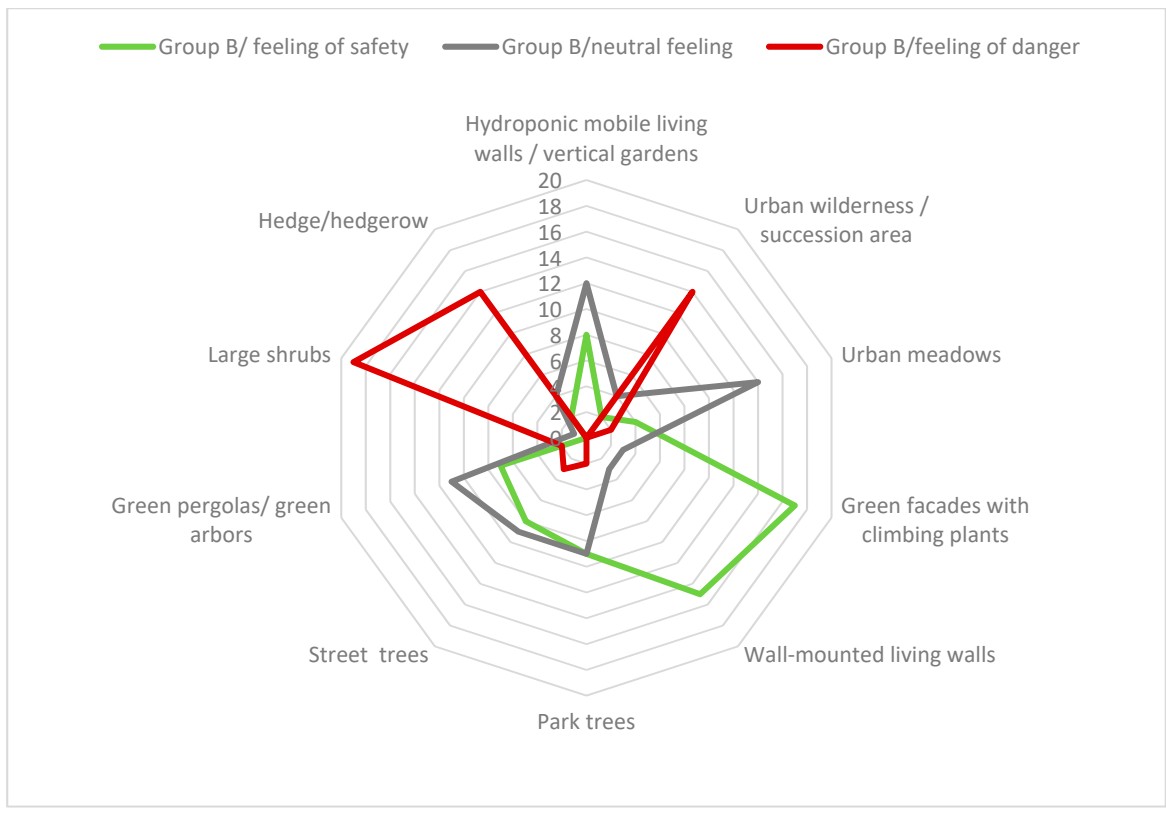

**Figure 9.** Radar charts with frequency of answers for selected NBS for the student group. Source: authors.

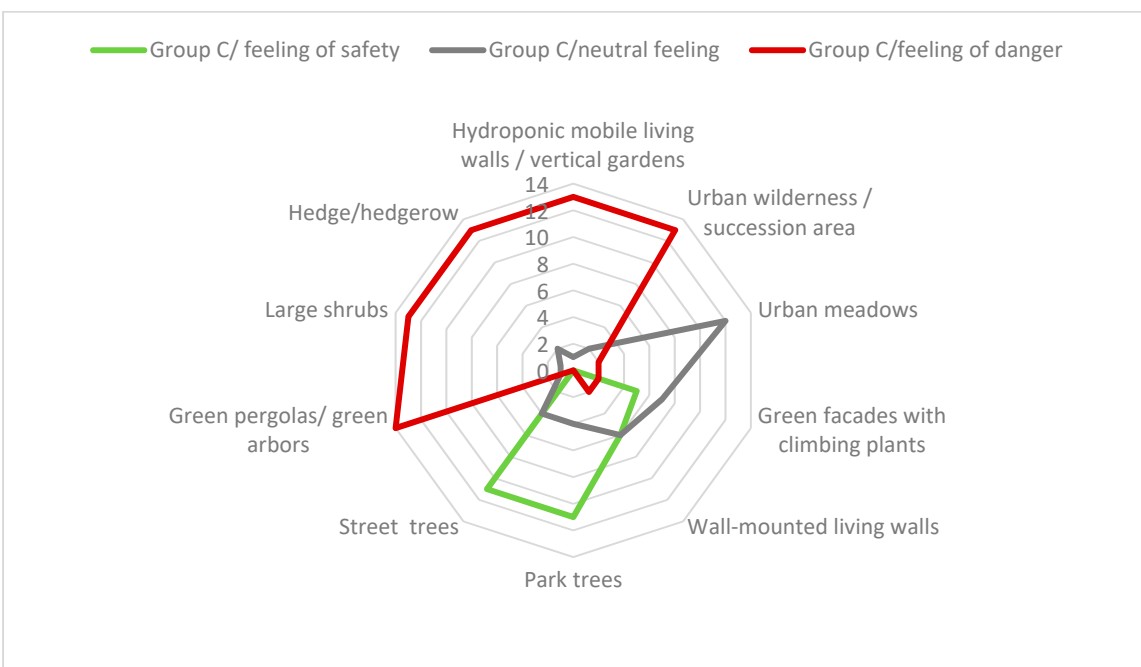

**Figure 10.** Radar charts with frequency of answers for selected NBS for the group from Přívoz Source: authors.

## 5. Discussion

The relationship between urban greenery and feelings of safety is not a new phenomenon [85]. The research shows that places that provide potential hiding places for perpetrators and limited visibility or escape are perceived as dangerous [48]. Large shrubs and urban wilderness can be considered such features. The result is therefore in accordance with the assumptions. Negative feelings about urban wilderness are also confirmed by [86].

Discussions can be held about the discrepancy between the research results and the already applied implementation of the NBS. An urban forest has been developed on brownfield sites in Leipzig with an obvious positive effect on the housing in the area by increasing the occupancy rate [53]. It can be assumed that the main difference is the embeddedness in the city structure. In Leipzig, these were former brownfields (size greater than 3 ha), which are now used for recreation but are a separate entity [53]. Residents are not forced to enter the forest area in their daily movements (possibly at night). In contrast, an area of small urban wilderness embedded in a public space that is used for the daily movement of residents is a place that residents must encounter in their daily movement. This is considered to be the main reason for the difference in residents' feelings.

Hedgerows were perceived differently. Respondents in Ostrava made general comments and did not indicate a specific location. Respondents from the Silesian agglomeration indicated a specific site and perceived shrubs more positively. The different approaches may be given by the very good experience of the Silesian agglomeration, the different types of housing development and especially the specific area.

As for the results, urban meadows are interesting, which were evaluated rather negatively compared to the usual assumptions or were not in direct contact with the current location. According to [87], urban meadows are perceived positively after they have been established, but [87] also admit that their positive impact on biodiversity needs to be explained to respondents, which confirms the rather positive attitude of respondents in the Silesian agglomeration. Biodiversity enhancement in low-mown urban grasslands is also addressed by [88], for example

In terms of vertical features, there were differences in results. Green facades with climbing plants and living walls had an average rating of less than two, meaning that most respondents considered them neutral or safe. This result was virtually confirmed in the

second pilot area. In contrast, elements of hydroponic furniture, living walls/vertical gardens, vertical vegetable/herb gardens and hanging wall planters (as green street furniture) were rated between 2 and 2.4. In the Silesian area, respondents were particularly positive about mobile and flexible walls. It can be assumed that there may have been a lack of perception of the robustness of the design used in the solution in Ostrava, for example, despite the fact that the functionality of such a system in cities has been a topic of an extensive expert discussion for a long time (e.g., [89]). But again, the good experience of the Silesian agglomeration may also have contributed to a different attitude by the respondents. In future research, more attention will need to be paid to vertical elements or should focus on them separately.

A significant limitation of the results for the Ostrava pilot is that there is only a three-level response scale. In further research, it will be necessary to adjust the scale. Respondents should express their feelings on a scale of 1–5 or 1–10. Another limitation that may have affected the result may have been photographs showing a particular element. For further research, it will be necessary to test the neutrality of the feeling evoked by the images in a sample of respondents.

A relatively small number of respondents can be considered a limitation to the research, especially in the Ostrava part. Given the number of respondents, the age composition does not match a demographic curve. Although equal numbers of men and women were approached, efforts to obtain gender-balanced responses were unsuccessful. It is therefore not possible to analyse gender differences in relation to feelings of threat from the data obtained.

A limitation of the research is the actual form of interviewing in the pilot area of Ostrava as mentioned in Chapter 3—Methods. As a result of the COVID-19 pandemic, the method of interviewing had to be modified. The initial assumption of face-to-face meetings in larger groups was not possible. Meeting with the senior generation was practically excluded for almost a year. Another limitation of the results that will need to be taken into account in future research is the likely changes in society as a result of the pandemic, which was a prolonged stay at home and changes in work and social habits.

## 6. Conclusions

Fear of crime is itself a problem that has costly and long-lasting consequences for a social life of a city, so understanding its causes and why it occurs as a social phenomenon plays a key role in developing the right policies [7].

The city as a living organism is a site of clashes of opinion between different groups of experts, even though they share the common goal of sustainable development. The paths that lead to this goal may be different.

This can be seen in the example of nature-based solutions. A little green acupuncture in the city can have a number of positive effects (reducing not only heat but also noise and dust). However, such a solution cannot be implemented everywhere where it is technically appropriate or advantageous. The acupuncture of an urban environment, especially when associated with brownfield issues, must respect the specificities of the site. Brownfields are often located in areas with higher crime rates. It is therefore necessary to consider and communicate well the development of such NBS that could create a sense of danger to citizens. For example, costly green walls or green roofs are positively received by residents in terms of being unsafe, while having a number of ecosystem services.

If the citizens do not have their own experience with NBS elements or the concept of applying them is not well communicated to them (as was the case in Ostrava—a resistant lecture is not sufficient), fears can arise, which can then lead to non-acceptance of these elements by the city residents.

Another possible explanation is its link to the imagination of the respondents. If they talked about a specific location, they had the opportunity to place a specific element in the context of the environment. However, if they were working with NBS without a predefined location their evaluation was influenced by their level of imagination and the place where

they probably subconsciously placed the element. However, this psychological aspect was not the focus of the study but may serve as a stimulus for further research in the field of urban psychology

This seemingly simple and clear and predictable result is important. In many cases, various elements of NBS are being built with the vision of creating sustainable and resilient cities. Society is falling into stereotypes that anything green is right and must be enthusiastically embraced by residents. Society is becoming significantly radicalized in environmental issues. Which can lead to a deeper dissatisfaction of a part of the population, for whom sustainable and resilient cities become uninhabitable and hostile.

**Author Contributions:** Conceptualization, B.V., I.T. and A.S.-S.; methodology, B.V., I.T. and A.S.-S.; validation, B.V., I.T. and A.S.-S.; formal analysis, B.V. and A.S.-S.; investigation, B.V., I.T. and A.S.-S.; resources, B.V. and I.T., data curation, B.V., I.T. and A.S.-S.; writing—original draft preparation, B.V., I.T. and A.S.-S.; writing—review and editing, B.V.; visualization, B.V. and A.S.-S.; supervision, B.V.; project administration, A.S.-S.; funding acquisition, A.S.-S. All authors have read and agreed to the published version of the manuscript.

**Funding:** This research was funded by SALUTE4CE—Integrated environmental management of SmALl Green Spots in FUncTional Urban ArEas following the idea of acupuncture, Project index number: CE1472, from Interreg CENTRAL EUROPE Programme.

**Institutional Review Board Statement:** Ethical review and approval were waived for this study due to REASON: The personal data that was provided by the respondents related only to age (age range) and gender. All questionnaires were fully anonymised. The questionnaires were processed only by the authors and the results of the individuals were not published or provided to any third party.

**Informed Consent Statement:** Not applicable.

**Data Availability Statement:** Not applicable.

**Acknowledgments:** This research was supported by the programme of the Polish Ministry of Science and Higher Education called: "PMW" in years 2019–2022 (contract no. 5062/INTERREG CE/19/202/2.

**Conflicts of Interest:** The authors declare no conflict of interest.

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
