# Peer review of "Implementing Nature-Based Solutions in Urban Spaces in the Context of the Sense of Danger That Citizens May Feel"

_land, doi:10.3390/land11101712_

Round 1
Reviewer 1 Report
It is my pleasure to review "Implementing Nature-based Solutions in Urban Spaces in the Context of the Sense of Danger that Citizens May Feel. The study describes The research revealed differences in respondents’ perception of the sense of danger when it comes to specific or theoretical locations. The research confirmed the need to involve citizens in the process of proposing the location of NBS in urban public spaces, thereby alleviating any potential worries." I have the following suggestions to improve this paper.
- Abstract
Please write sentences that can mention the problem statement and then initiate this paper.
- Introduction
There is a lack of a recent literature review on the current topic.
The authors need to more clearly state and show in the introduction of the paper why and how it is different from previous methodologies and why this work is of advanced knowledge.
- Materials and Methods:
The lack of details i.e. Participants' age, and other multiple factors could impact the results.
How do the authors accommodate the different things of each Participant which could impact the results?
The question should be more for a comprehensive study and for a scientific publication.
- Results and discussion
It is better to explain the result in detail as well as add 1-2 more figures from your results analysis. In the present form, there is a need to be more work on the presentation of work.
I think it is better to explain more details about your results.
- Conclusion
What are the limitations of the current research?
The English level is generally good but should be edited for simplification and clarity. Shorter sentences and a focus on key points in the methods and results, in particular, would benefit the communication of the work.
Best,
Author Response
Thank you very much for taking the time to read our article. Thank you also for your review and helpful recommendations.
We have tried to improve the whole text based on your suggestions - the added sentences are marked in blue.
We have completely changed the abstract in the text. I have also added information to the introduction. We removed Figure 3 and added additional figures showing the results. I have added a discussion section. We also added a conclusion. We hope that we have managed to make the article acceptable for publication.
We submitted the article for proofreading and then corrected it.
Reviewer 2 Report
After reading the submitted text, my comments can be summarized in a few sentences. What is tested? The threat posed to people who use construction elements / land use (as far as this text is concerned)? Discomfort in using this space? The impact of discomfort on the next? Planning to plan / use appropriate site usage conditions? Fear of crime (talk 456 to 458 about something)? I admit that I got lost in it ... How to investigate? There is a description of individual groups and their size (the authors note that the sample is very small and has some limitations). But nothing has emerged about the method of selecting a random sample for the survey, the latter is probably purposeful) and when this sample is to be overflowed? How to generalize the conclusions in this case? Thus, from the context it follows that he did (a) linking with research in the case of Poland - one-time, in the case of the Czech Republic - on unrelated - outlays on himself, as well as in this case, in this case, approaches to be done? Can you compare them with each other? What's so on the course to school? A method / model of shaping space with the use of NBS? Do you consider negative reception of wild or grassy substitute justifications by, for example, living walls and, wherever possible, climbing plants? We that these spaces create a different environment and people's feelings are not transferred to other forms of these forms / use, how well to read the writings of the authors, which means that the writings are respected. Perhaps, then, to start swimming in terms of information action of soft flies / reducing them and their associations? Additionally, in the text the inscriptions 'we', 'our' or other are repeated many times [lines 13, 66, 151, 131, 336, 440, 445] should be written in a personal form. I take it for the very end. What was the interpolation method used to derive Figure 3? And as for security and changes that may make an additional change or two guaranteed based on scientific research that is in line with the security guarantee. In my opinion, the short information from 125-127 lines should be expanded and based on the literature on the subject.
Author Response
Thank you very much for taking the time to read our article. Thank you also for your review and helpful recommendations.
We have tried to improve the whole text based on your suggestions - the added sentences are marked in blue.
We tested the feelings of the respondents. Their fears. We did not break down these fears further in this research. In the next steps, we consider to divide the fears into fears of crime, fears of technical failure, fears of insects, birds. Concerns may also be about allergies.
In the text, I have tried to better explain the sampling of respondents. We hope that we have managed to better.
We understand that the adoption of different elements in an urban space can provoke different reactions. Some changes are better received and some are less well received. Also, society is not always homogeneous. We are aware of all this. But we believe that it is necessary to approach the application of even environmentally beneficial elements responsibly, after discussion, not only with experts but also with citizens. The aim is to create a pleasant living environment. At the same time, however, we also need to understand the limits of the respondents in terms of imagination or assessment of long-term wishes.
We have completely changed the abstract in the text. I have also added information to the introduction. Removed Figure 3 (was add only fr ilustration) and added additional figures showing the results. We have added a discussion section. We also added a conclusion.
Thank you also for your comment on the overly personal description. We tried to edit the article.
We hope that we have managed to make the paper acceptable for publication.
We submitted the article for proofreading and then corrected it.
Reviewer 3 Report
The paper is apt for this Journal as it assesses the sense of danger in relation to some NBS and revealed differences in respondents’ perception of the sense of danger when it comes to specific or theoretical locations.
The case study analysis is suitable for publication, while the first part is still not good enough in the first part (introduction). Here a more reasoning is necessary on theoretical framework /literature review.
I highly suggest to clarify the relationship between NBS and brownfield by taking into consideration these themes and related references:
- https://www.elgaronline.com/view/book/9781800375611/book-part-9781800375611-18.xml, which especially focus on green infrastructure
- https://www.mdpi.com/2073-445X/10/9/893 on afforestation;
-https://www.sciencedirect.com/science/article/abs/pii/S0048969719303985, on NBS for contaminated land remediation and brownfield redevelopment in cities
- https://www.tandfonline.com/doi/full/10.1080/14649357.2016.1158907 on NBS and sustainable cities
- https://journals.sagepub.com/doi/10.1177/00420980211045571, https://www.nature.com/articles/s41599-022-01145-0, on the increasing need to provide cities with NBS and more human-centered environments, further pushed by the pandemic
- (2021). The Green City and Social Injustice. 21 tales from North America and Europe. Routledge on green cities
So, please, enrich the introduction by dealing with these themes with the suggested references.
I am looking forward for the new version of the paper.
Author Response
Thank you very much for taking the time to read our article. Thank you also for your review and helpful recommendations.
We have tried to improve the whole text based on your suggestions - the added sentences are marked in blue.
We would especially like to thank you for the very useful references to the literature. Unfortunately, we were not able to find all publications. Nevertheless, we have tried to improve the introduction and the discussion
We have completely changed the abstract in the text. I have also added information to the introduction. Removed Figure 3 and added additional figures showing the results. I have added a discussion section. We also added a conclusion. We hope that we have managed to make the paper acceptable for publication.
We submitted the article for proofreading and then corrected it.
Round 2
Reviewer 1 Report
Some of the comments which are not answered:
1. The lack of details i.e. Participants' age, and other multiple factors could impact the results. How do the authors accommodate the different things of each Participant which could impact the results?
2. Figure 4 should be in English.
3. It is better to explain the result in detail as well as add 1-2 more figures from your results analysis. In the present form, there is a need to be more work on the presentation of work. analysis. In the present form, there is a need to be more work on the presentation of work.
Best,
Author Response
Thank you again for your patience. Also, thank you for your stimulating comments.
All changes are written in change mode and are visible in the text
Based on the comment, we have added the age composition (starting at line 260.)
In connection with this, we have added 3 graphs of response rates (Fig 7-9).
We have added an English translation to Figure 3.
We hope that we have been able to meet your requirements. We really tried to fulfill especially point 3 .
Reviewer 2 Report
Most of my previous comments have been positively verified. I have no major comments to the text sent. Except for one, maybe. There are still personal forms in the text [verse 475, 480, 503]. The text should be written in an impersonal form. This is relatively easy to fix in the next revision of the article.
Author Response
Thank you for your patience with our article.
We have tried to improve it further based on the requests of rewiers, also we have tried to exclude all personal forms. We have added the text of the article in change mode so that the individual changes are visible. We hope that have been more successful this time. We hopethat have managed to edit the article to your satisfaction.
Reviewer 3 Report
The paper has certainly improved, but still lacks of three points:
1) the dimension of planning for green infrastructure on the basis of brownfield infrastructure (see https://www.elgaronline.com/view/book/9781800375611/book-part-9781800375611-18.xml);
2) In connection to this, NBS for contaminated land remediation and brownfield redevelopment in cities (https://www.sciencedirect.com/science/article/abs/pii/S0048969719303985); and
3) Reflection on social justice implication on the green city, (2021). The Green City and Social Injustice. 21 tales from North America and Europe. Routledge on green cities.
Author Response
Thank you for your patience with our article.
Thank you very much for sending us the texts, they helped us a lot. We have added the text of the article in change mode so that the individual changes are visible. We have added references to the recommended literature. While for the recommended book we have also referred separately to the individual chapters. However, we find the whole book ( Green city) very stimulating and will certainly help us in our future work. Thank you again for your supportive comments.